# Buffering Mitigates Chondrocyte Oxidative Stress, Metabolic Dysfunction, and Death Induced by Normal Saline: Formulation of a Novel Arthroscopic Irrigant

**DOI:** 10.3390/ijms25021286

**Published:** 2024-01-20

**Authors:** Arman Hlas, Venkateswaran Ganesh, Jaison Marks, Rui He, Aliasger K. Salem, Joseph A. Buckwalter, Kyle R. Duchman, Kyungsup Shin, James A. Martin, Dongrim Seol

**Affiliations:** 1Carver College of Medicine, University of Iowa, Iowa City, IA 52242, USA; arman-hlas@uiowa.edu; 2Department of Orthopedics and Rehabilitation, Carver College of Medicine, University of Iowa, Iowa City, IA 52242, USA; venkateswaran-ganesh@uiowa.edu (V.G.); jaison-marks@uiowa.edu (J.M.); joseph-buckwalter@uiowa.edu (J.A.B.); kyle-duchman@uiowa.edu (K.R.D.); 3Roy J. Carver Department of Biomedical Engineering, College of Engineering, University of Iowa, Iowa City, IA 52242, USA; 4Pharmaceutical Sciences and Experimental Therapeutics, College of Pharmacy, University of Iowa, Iowa City, IA 52242, USA; rui-he@uiowa.edu (R.H.); aliasger-salem@uiowa.edu (A.K.S.); 5Department of Orthodontics, College of Dentistry and Dental Clinics, University of Iowa, Iowa City, IA 52242, USA; kyungsup-shin@uiowa.edu

**Keywords:** arthroscopic irrigation, normal saline, acidity, oxidative stress, metabolic dysfunction, cell death, HEPES, reactive oxygen species, chondrocyte, synoviocyte

## Abstract

For decades, surgeons have utilized 0.9% normal saline (NS) for joint irrigation to improve visualization during arthroscopic procedures. This continues despite mounting evidence that NS exposure impairs chondrocyte metabolism and compromises articular cartilage function. We hypothesized that chondrocyte oxidative stress induced by low pH is the dominant factor driving NS toxicity, and that buffering NS to increase its pH would mitigate these effects. Effects on chondrocyte viability, reactive oxygen species (ROS) production, and overall metabolic function were assessed. Even brief exposure to NS caused cell death, ROS overproduction, and disruption of glycolysis, pentose phosphate, and tricarboxylic acid (TCA) cycle pathways. NS also stimulated ROS overproduction in synovial cells that could adversely alter the synovial function and subsequently the entire joint health. Buffering NS with 25 mM 4-(2-hydroxyethyl)-1-piperazineethanesulfonic acid (HEPES) significantly increased chondrocyte viability, reduced ROS production, and returned metabolite levels to near control levels while also reducing ROS production in synovial cells. These results confirm that chondrocytes and synoviocytes are vulnerable to insult from the acidic pH of NS and demonstrate that adding a buffering agent to NS averts many of its most harmful effects.

## 1. Introduction

Arthroscopy is a widely utilized, minimally invasive surgical technique used for diagnosing and treating joint pathologies in an expanding number of joints. Visualization during arthroscopy is improved by continuous fluid irrigation, which provides joint distension, maintenance of hydrostatic pressure, and clearance of joint debris [1]. While patients benefit from expedited recovery, reduced postoperative swelling, and a lower risk of complications compared to open arthrotomy [2], irrigation solution has the potential to damage articular cartilage due in part to the sub-optimal composition of 0.9% normal saline (NS), the most commonly used irrigant [3,4].

NS can create an unfavorable environment for chondrocytes due to its acidity [5,6,7]. Acidity causes oxidative stress and mitochondrial dysfunction, leading to decreased intracellular adenosine triphosphate (ATP) and mitochondrial membrane potential, while inducing matrix metalloprotease (MMP) expression and subsequent proteoglycan depletion [8]. Additional studies have revealed that NS exposure reduced radiolabeled sulfate uptake, indicating inhibition of proteoglycan synthesis, as well as increased coefficient of friction, an indicator of lubricin depletion from cartilage surfaces [3,4,9]. These effects may lead to cartilage degeneration and osteoarthritis. Despite these well-studied detrimental effects, NS is still commonly used and little progress has been made in adopting a more biocompatible irrigant.

Many irrigation fluids including Ringer’s solution/lactate, mannitol, sorbitol, dextrose, dextran, and glycine have been introduced as superior to NS in terms of ionicity, osmolarity, nutrition, and pH [10]. In particular, chondrocyte metabolism was remarkably reduced with excursions of as little as pH 0.2 units from neutral [11]. Moreover, it is known that chondrocyte exposure to acidic media induces increases in oxidative stress markers [8]. Therefore, buffering NS to neutralize its pH would mitigate cellular oxidative stress.

In this study, we sought to better define the mechanisms of NS toxicity in an effort to develop a practical intervention that can be rapidly and widely adopted for clinical use. We found that simply adjusting the pH of NS with 4-(2-hydroxyethyl)-1-piperazineethanesulfonic acid (HEPES), a biocompatible buffering agent, profoundly moderated the chondrocyte oxidative and metabolic damage caused by unbuffered NS.

## 2. Results

Figure 1 shows the cytotoxicity of 0.9% NS and buffers verified by viability, death, and morphologic analyses. The decrease in viability of bovine chondrocytes depended on the exposure time to NS: 92.8 ± 7.8% at 2 h (h) (*p* = 0.586 vs. control) and 43.5 ± 13.9% at 3 h (*p* < 0.001 vs. control) (Figure 1a). The toxicity of NS was validated by flow cytometry analysis and morphological changes. In flow cytometry, the healthy cell population in control was approximately 95% with almost no sign of apoptosis or necrosis; however, NS exposure induced 15.2% and 99.4% cell death at 30 min (min) and 3 h (Figure 1b and Appendix A). In phalloidin staining, the NS treatment induced severe cytoskeletal damage of chondrocytes, resulting in dramatic cell shrinkage and detachment (Figure 1c). Moreover, we could observe a large population of cell debris when treated with the NS group (Appendix A).

As well-known buffering agents to maintain pH, 25 mM HEPES or 0.5 mM sodium bicarbonate were added in NS. The pH level of culture media as a control was within physiological range (7.85 ± 0.009), whereas 0.9% NS recorded an acidic pH of 5.30 ± 0.017 (Table 1). There was no change in intracellular sodium after NS exposure after 30 min (Appendix A). Supplementation of HEPES or sodium bicarbonate in 0.9% NS increased the pH to 7.05 ± 0.005 or 7.30 ± 0.012, respectively. Interestingly, there was a minor pH change during the exposure of control and HEPES with chondrocytes, while the pH of NS and sodium bicarbonate buffer became 9.6% and 10.3% more acidic at 3 h, respectively. The effect of pH buffering was promising to reduce chondrocyte death. The rates of necrotic and late apoptotic cells were significantly diminished in both HEPES and sodium bicarbonate groups; however, the difference was notable at 3 h exposure (Annexin V/PI positive: 7.7% in HEPES vs. 52.6% in sodium bicarbonate) (Figure 1b and Appendix A). Therefore, only the HEPES supplement was used for further studies. Phalloidin staining revealed the protection of cellular morphology when NS was supplemented with HEPES (Figure 1c).

Chondrocytes treated with NS had an increased release of reactive oxygen species (ROS), as imaged by the DHE staining (Figure 2). The levels of ROS after 3 h of NS exposure were approximately 500-fold (*p* = 0.03 vs. control) and 2.4-fold (*p* < 0.001 vs. control) higher than control in 2D monolayer (Figure 2a,b) and 3D agarose culture systems (Figure 2c,d), respectively. In monolayer cultures, the addition of HEPES significantly reduced ROS levels (*p* = 0.033 vs. NS) (Figure 2a,b). Similarly, the DHE staining of chondrocytes encapsulated in agarose hydrogel was significantly decreased with HEPES addition (*p* = 0.012 vs. NS) (Figure 2c,d). In addition to chondrocytes, bovine synoviocytes exposed to NS showed a similar trend of ROS overproduction (Figure 3). The NS induced a significant 3-fold increase in ROS level by synoviocytes (*p* = 0.002 vs. control), while the addition of 25 mM HEPES supplementation negated this deleterious effect (*p* < 0.001 vs. NS).

At least 2-fold up/down-expressed metabolites are listed in Appendix A. Exploratory statistical analysis of the top 40 metabolites displayed in the heatmap with hierarchical clustering showed a clear divergence between control and NS as well as between NS and NS+HEPES (Figure 4). HEPES supplementation reduced the extent of alterations induced by NS in the glycolytic, pentose phosphate, and tricarboxylic acid (TCA) cycle pathways. NS induced an increase in intermediates in the glycolysis pathway (glucose 6-phosphate and fructose 6-phosphate) (Figure 5b and Figure 5c, respectively), whereas the levels of the final products, pyruvate and adenosine triphosphate (ATP), were significantly decreased (*p* < 0.001 vs. control) (Figure 5f and Figure 5g, respectively), an effect that HEPES mitigated (*p* < 0.001 vs. NS) (Figure 5f and Figure 5g, respectively). HEPES also significantly moderated the suppressive effects of NS on the levels of dihydroxyacetone phosphate (DHAP) and 3-phosphoglycerate (3PG) (Figure 5d and Figure 5e, respectively). NS exposure activated the pentose phosphate pathway (PPP) with a higher expression of ribulose 5-phosphate (Ru5P) and ribose 5-phosphate (R5P) (*p* < 0.001 vs. control) (Figure 5j and Figure 5k, respectively). Reduced nicotinamide adenine dinucleotide phosphate (NADPH) in the NS group was approximately 31-fold lower than in the control group (Figure 5i). HEPES addition induced full recovery of reduced nicotinamide adenine dinucleotide (NADH) and guanosine triphosphate (GTP) (*p* < 0.001 vs. control) (Figure 5p and Figure 5q, respectively). Lastly, NS caused a 7.2-fold reduction of lactate production (*p* < 0.001 vs. control) (Figure 5r). In contrast, levels were close to control in the NS+HEPES. TCA metabolites, including acetyl coenzyme A (acetyl-CoA) (Figure 5l), α-ketoglutarate (α-KG) (Figure 5m), fumarate (Figure 5n), and malate (Figure 5o), were significantly decreased relative to control in both the NS and NS+HEPES groups.

## 3. Discussion

Although NS irrigation is commonly used to aid visualization during arthroscopic procedures, several studies have reported that it results in chondrocyte death and inhibits cartilage metabolism [3,4,6,12]. In the present study, we hypothesized that such adverse effects are derived from the acidic pH of NS. The NS exposure induced ROS production in chondrocytes resulting in a time-dependent cell shrinkage, detachment, and death without any intervention (Figure 1 and Figure 2). Similar oxidative damage was also observed in synoviocytes, a key cell population of the intra-articular space (Figure 3). In addition, metabolomics analysis evidenced the profoundly disrupted key metabolic pathways related to energy production, intra-cellular signaling, and antioxidant defenses (Figure 4 and Figure 5). The addition of HEPES to NS as a buffering agent significantly reduced ROS production and cytotoxicity, supporting the hypothesis that the sub-physiologic pH of NS is at least partly to blame for its negative effects on chondrocytes.

In cartilage, chondrocytes maintain a superficial to deep zone pH gradient of 6.9 to 7.2, and a reduction of as little as 5% significantly impairs energy production and matrix synthesis [11]. Metabolomics analysis after NS exposure clearly delineated the dysregulation of pathways contributing to chondrocyte energy production. (Figure 4 and Figure 5). The near 7-fold decrease in lactate levels after NS exposure is consistent with glycolytic pathway impairment. Buffering NS with the HEPES addition jointly highlights the inhibition of anaerobic glycolysis, a commonly preferred energy mechanism for chondrocytes under low pH conditions.

Short-term exposure to NS clearly suppressed the ATP, GTP, NADH, and NADPH levels (Figure 5). The lowered ATP-to-ADP ratio signals compromise cellular energy mechanisms, leading to an increase in AMP that signals the need for glycolysis to address energy needs. Unlike ATP, which is the central energy currency of cellular functions, GTP addresses limited needs, such as signal transduction and protein synthesis. The similar pathologic trend in GTP-to-GDP ratio from NS treatment and HEPES-supplemented recovery of the same clearly signals damage of the associated pathways from acidic pH. NADPH:NADP+ ratio in the NS-treated group decreased by nearly 100-fold versus control, but HEPES supplementation restored the attenuation by nearly 80%. Another notable change after NS exposure was an approximately 65% loss of pyruvate, which is an important metabolite to maintain cell homeostasis and to suppress ROS generation in the mitochondria. In addition to pyruvate, the expression of several metabolites regulating antioxidation such as α-KG [13] and β-hydroxybutyrate [14] were significantly reduced after NS exposure.

Collins et al. reported that human chondrocytes incubated in acidic culture media (pH 6.2) at 5% O_2_ showed several markers of oxidative stress such as an increased ROS generation, decreased mitochondrial membrane potential, and a reduced/oxidized glutathione (GSH/GSSG) ratio [8]. Our oxidative stress data supports the deleterious effect of low pH on both chondrocytes and synoviocytes, while pH adjustment of NS by HEPES supplementation reduced the oxidative stress (Figure 2 and Figure 3). This explains the earlier observed role of HEPES in maintaining the activities of the glycolytic and pentose phosphate pathways at near control levels, as both are required for proteoglycan synthesis. HEPES is an organic and biologically compatible buffer that is widely used in cell culture at 10–25 mM to maintain physiological pH. We additionally prepared the most common buffer used in mammalian cell culture, 0.5 mM sodium bicarbonate, to validate if pH adjustment is the critical factor. In the flow cytometry study with Annexin V/PI staining, the rate of chondrocyte death was remarkably reduced until 30 min, when the pH was neutralized with sodium bicarbonate (Figure 1b and Appendix A). However, the protective effect of sodium bicarbonate supplement on cell death was relatively diminished at 3 h of exposure compared to HEPES. Thus, pH buffering can be a major factor in maintaining cellular integrity, but further studies are needed to understand the superior mechanism of HEPES. Besides pH buffers, several candidate irrigation supplements were tested for the effects on ROS production via DHE staining (Appendix A). While glucose and mannitol supplementation had no significant effect on DHE staining and cell death when compared with NS alone, the commonly used antioxidant *N*-acetylcysteine (NAC) supplementation was as chondroprotective as HEPES, linking its buffering capacity to oxidant protection [15].

While the goal of arthroscopy remains to correct articular pathologies and restore overall joint function, these procedures are not without risk of additional injury. Compton et al. found the rate of iatrogenic cartilage injury in academic training videos to be 73.8%, with minor simulated iatrogenic injury leading to at least some degree of chondrocyte death [16]. Teeple et al. found arthroscopic irrigation in the superficial zone of bovine stifle joints enhanced the loss of lubricin, an essential boundary lubricant protecting cartilage from mechanical damage to articular chondrocyte surfaces [17]. Our results additionally suggest that NS exposure itself may cause further injury to chondrocytes and synovium in an already damaged joint. The cumulative effects of this damage may be difficult to quantify in the immediate postoperative period; however, these adverse effects may cause delayed recovery or contribute to an increased risk of osteoarthritis and potential arthroplasty in patients who have previously undergone arthroscopy [18].

Although we mainly focused on articular cartilage in this study, the synovium has a critical role in affecting the joint environment. Since the tissue consists of two to three layers of synoviocytes in the intima, it is highly susceptible to fluctuations in the pH that is associated with joint irrigation [19]. In the present study, NS exposure induced the elevation of ROS levels in synoviocytes (Figure 3). While further investigation is needed, synovial damage via oxidative stress may contribute to post-operative joint swelling, stiffness, pain, and/or osteoarthritis if not properly treated [15,20].

We used two different in vitro cell culture models: (1) 2D monolayer culture to evaluate toxicity, morphology, and metabolomics, and (2) 3D agarose hydrogel cultures to assess oxidant production, as it mimics a native tissue to maintain cell morphology and phenotype by promoting adjacent cell interactions [21]. Although agarose hydrogel enables a 3D microenvironment for the chondrocytes, it does not truly capture the zonal characteristics of a native tissue, which is likely to affect NS penetration and ensue changes in the pH gradient. Therefore, the deleterious effects of NS we observed here should be validated in an animal study. Another limitation in this study was that we used a static culture condition. During arthroscopic operation, the mechanically circulated irrigant with fluid pressure can induce structural damage to the cartilage surface [17]. Thus, the oxidative damage and metabolic changes in the NS-treated cells may be accelerated under dynamic fluid flow conditions.

## 4. Materials and Methods

### 4.1. Tissue Harvest and Cell Isolation

Bovine stifle joints of freshly slaughtered young adult cattle (15–24 months old) were procured from a local butcher store (Bud’s Custom Meats, Riverside, IA, USA), and articular cartilage strips were harvested under sterile conditions. Following washing in Hanks’ Balanced Salt Solution (HBSS) (Life Technologies, Carlsbad, CA, USA) containing 100 U/mL penicillin-streptomycin (Life Technologies) and 2.5 μg/mL amphotericin B (Life Technologies), primary chondrocytes were isolated by a digestion method with a mixture of 0.25 mg/mL type-1 collagenase (Sigma-Aldrich, St. Louis, MO, USA) and protease (Sigma-Aldrich). Synoviocytes were isolated from the inner joint capsule (synovial intima) by gentle scraping. Both cells were cultured in 50:50 Dulbecco’s modified Eagle’s medium and Ham’s F-12 Nutrient Mixture (DMEM/F-12; Life Technologies) supplemented with 10% fetal bovine serum (Life Technologies) and antibiotics in hypoxic culture condition (5% O_2_/CO_2_ at 37 °C). Early passages (P1–3) of cells were used for all data acquisition here.

### 4.2. Two-Dimensional (2D) Monolayer and Three-Dimensional (3D) Agarose Gel Culture Systems

Monolayer and 3D agarose cultures were used to investigate the effects of irrigants on cell viability, morphology, and oxidant production using fluorescent probes and confocal microscopy. For 3D culture, a 4% (*w*/*v*) stock solution of low-gelling agarose (Research Products International, Mount Prospect, IL, USA) was prepared in double-distilled water and mixed 1:1 with cell suspensions. Final concentrations of chondrocytes and synoviocytes were 5 × 10^6^ cells/mL and 2 × 10^6^ cells/mL, respectively, in 2% (*w*/*v*) agarose gel. The mixtures were plated in a 6-well plate and pre-cultured for 3 days prior to the incubation of irrigation solutions.

### 4.3. Irrigation Solutions

Sterilized 0.9% NS were purchased from Baxter (Deerfield, IL, USA), and culture media (DMEM/F-12 with supplements) served as a control for all experiments. HEPES (Life Technologies) as a zwitterionic sulfonic acid buffering agent and sodium bicarbonate (NaHCO_3_; Research Products International) were added to the NS for final concentrations of 25 mM and 0.5 mM, respectively. The pH scales for all 4 irrigation solutions were measured using a pH meter (Accumet^TM^ AB315, Thermo Fisher Scientific, Waltham, MA, USA) before and after exposure to chondrocytes. All solutions were incubated for up to 3 h in a static culture condition.

The amounts of intracellular sodium were measured using a cell-permeant sodium fluorescent indicator (SBFI-AM, Santa Cruz Biotechnology, Dallas, TX, USA) according to the manufacturer’s instruction. Briefly, bovine chondrocytes were exposed with culture media (control), 0.9% NS, 25 mM HEPES, or 0.5 mM sodium bicarbonate (NaHCO_3_) for 30 min, and then 10 µM SBFI-AM dissolved in anhydrous dimethyl sulfoxide (Biotium, Fremont, CA, USA) mixed with 0.1% (*w*/*v*) F-127 Pluronic acid (Millipore Sigma) was added. After 1 h incubation, the samples were gently centrifuged (50× *g* for 30 s (s)) and measured for fluorescence using a SpectraMax Plus spectrophotometer (Molecular Devices, San Jose, CA, USA). The raw data were normalized by the control.

### 4.4. Cell Viability

To assess the extent of cell viability in the monolayer culture, primary bovine chondrocytes were plated in a 96-well plate and incubated in culture media (control) for 3 h or 0.9% NS for 0.5, 1, 2, and 3 h. The cells were washed after the treatment, and the viability was determined by CellTiter 96^®^Aqueous One Solution (Promega, Madison, WI, USA). An amount of 20 µL of One Solution Reagent was added into each well containing 100 µL media, and the plate was incubated at 37 °C in a humidified incubator for 1.5 h. The absorbance was recorded at 490 nm using the microplate spectrophotometer.

The viability of chondrocytes treated with NS and buffers was intensively assessed using the CF488A-Annexin V and PI apoptosis assay kit (Biotium, Fremont, CA, USA) according to the manufacturer’s protocol. Briefly, bovine chondrocytes were incubated with irrigation solutions for 0.5, 1, and 3 h, and both the supernatant and the adherent cells were collected. Subsequently, the cells were washed with Dulbecco’s phosphate buffer solution (DPBS), centrifuged, and resuspended in the binding buffer provided by the kit. The cells were then incubated in the dark, on ice, with or without 5 µL of CF^®^488A Annexin V and/or 1 µL of PI working solution, for 20 min. Flow cytometry was performed using a FACSCalibur flow cytometer (BD Biosciences, San Jose, CA, USA) and results were analyzed by FlowJo v10 (BD Biosciences). The flow cytometry plots were divided into four regions labeled Q1 (Annexin V negative/PI positive), Q2 (Annexin V positive/PI positive: necrotic and late apoptotic cells), Q3 (Annexin V positive/PI negative: early apoptotic cells), and Q4 (Annexin V negative/PI negative: healthy cells) (Appendix A).

### 4.5. Visualization of Cellular Morphology

After incubation with irrigation solutions, the cellular morphologic changes were verified by phalloidin immunofluorescence staining to label F-actin of the cytoskeleton. Chondrocytes plated in 8-chamber slides were fixed with 4% paraformaldehyde at room temperature (RT) for 10 min to preserve cytoskeletal structures. The cells were rinsed with cold phosphate buffer solution (PBS) (Life Technologies) thrice, permeabilized with 0.1% Triton X-100 at RT for 5 min, and treated with 1% bovine serum albumin (Research Product International) as a blocking agent at RT for 30 min. One unit of AlexaFlour-488 conjugated phalloidin methanolic stock solution (Life Technologies) was added to 200 μL blocking solution per chamber and continued incubation at 4 °C up to 1 h. Post-rinsing thrice with cold PBS, the stained slides were cover-slipped with 4′,6-diamidino-2-phenylindole (DAPI)-mounting media (VECTASHIELD^®^, Vector Laboratories, Newark, CA, USA) prior to confocal imaging using an Olympus FV1000 confocal microscope (Olympus, Center Valley, PA, USA).

### 4.6. Oxidative Stress Markers

Following experimental incubations, 6-well plates with monolayer cells and the cell-laden agarose hydrogel constructs were co-stained with Calcein AM (1:1000 dilution; green color; Thermo Fisher Scientific) and dihydroethidium (DHE; 1:500 dilution; red color; Thermo Fisher Scientific) to visualize live cells and ROS production, respectively, in cell culture media for 40 min. The cells were washed and imaged by an Olympus FV1000 confocal microscope. The fluorescence images of Calcein AM and DHE were processed to quantify the area via ImageJ 1.49p (NIH, Bethesda, MD, USA), and the mean values were normalized to the control group.

### 4.7. Metabolomics

In total, three experimental groups including untreated control, 0.9% NS, and 25 mM HEPES supplementation in NS were cultured in 6-well plates with 6 replicates per group. After exposure, the chondrocytes were washed with ice-cold PBS and ultrapure water, twice. The cells were gently exposed to the gaseous phase for 10–15 s before submerging into the liquid nitrogen for up to 2 min. The frozen cells were lyophilized overnight and then scraped into 1 mL of ice-cold methanol/acetonitrile/water (2:2:1 *v*/*v*) containing an internal standard (D4-citric acid, D4-succinic acid, D8-valine, and U13C-labeled glutamine, glutamic acid, lysine, methionine, serine, and tryptophan) (Cambridge Isotope Laboratories, Tewksbury, MA, USA) to extract metabolites. The mixtures were frozen in liquid nitrogen, thawed for 10 min, and rotated at −20 °C for 1 h. After centrifugation (21,000× *g* for 10 min), 300 µL of the cleared metabolite extracts were transferred to autosampler vials and dried using a SpeedVac vacuum concentrator (Thermo Fisher Scientific).

Metabolome was profiled using high-resolution mass spectrometry, following gas and liquid chromatography (GC-MS and LC-MS), in the Metabolomics Core Facility of the University of Iowa. For GC-MS, dried metabolite extracts were reconstituted in 20 µL of 11.4 mg/mL methoxyamine (MOX) in anhydrous pyridine, vortexed for 5 min, and heated at 60 °C for 1 h. Then, the samples were added with 16 μL N,O-Bis(trimethylsilyl)trifluoroacetamide (TMS), vortexed for 1 min, and heated at 60 °C for 30 min. Derivatized samples (1 µL) were injected into a Trace 1300 GC (Thermo Fisher) fitted with a TraceGold TG-5SilMS column (Thermo Fisher) operating under the following conditions: split ratio = 20:1, split flow = 24 µL/min, purge flow = 5 mL/min, carrier mode = constant flow, and carrier flow rate = 1.2 mL/min. The GC oven temperature gradient was as follows: 80 °C for 3 min, increasing at a rate of 20 °C/min to 280 °C, and holding at a temperature at 280 °C for 8 min. Ion detection was performed by an ISQ 7000 mass spectrometer (Thermo Fisher) operated from 3.9 to 21.0 min in EI mode (−70 eV) using select ion monitoring (SIM). For LC-MS, dried extracts were reconstituted in 20 µL acetonitrile/water (1:1 *v*/*v*), rotated at −20 °C overnight, centrifuged, and transferred to autosampler vials for analysis. Two µL of derivatized sample was injected into a Q Exactive hybrid quadrupole Orbitrap mass spectrometer (Thermo Fisher) with a Vanquish Flex UHPLC system (Thermo Fisher) fitted with a SeQuant ZIC-pHILIC column (2.1 × 150 mm, 5 µm particle size; Millipore Sigma, Burlington, MA, USA) with a ZIC-pHILIC guard column (20 × 2.1 mm; Millipore Sigma). Two solvents were utilized for the mobile phase. Solvent A consisted of 20 mM ammonium carbonate and 0.1% (*v*/*v*) ammonium hydroxide having a pH approximating 9.1. Solvent B consisted of acetonitrile. The method was run at a flow rate of 0.15 mL/min. The gradient started at 80% B and decreased to 20% B over 20 min; returned to 80% B in 0.5 min; and was held there for 7 min. The MS was operated in full-scan: polarity-switching mode from 1 to 20 min, 3.0 kV spray voltage, at 275 °C heated capillary, and 350 °C HESI probe. The sheath, auxiliary gas flow, and sweep gas flow were set to 40 units, 15 units, and 1 unit, respectively. MS data acquisition was performed in a range of *m*/*z* 70–1000 with the resolution set at 70,000, the AGC target at 1 × 10^6^, and the maximum injection time at 200 ms.

Raw data from metabolomics were analyzed using TraceFinder 5.1 (Thermo Fisher). A pooled sample generated prior to derivatization was analyzed at the beginning, at a set interval during, and at the end of the analytical run to correct peak intensities using the normalization and evaluation of MS-based metabolomics data (NOREVA) tool [22]. The NOREVA corrected data were then normalized to the total signal per sample to control for extraction, derivatization, and/or loading effects. Corrected data was then uploaded to MetaboAnalyst 5.0 for the heatmap analysis with hierarchical clustering.

### 4.8. Statistics

All quantified data were normalized to control (culture media). Mean values and standard deviations were shown in the bar graphs. Data were analyzed by one-way ANOVA with the Tukey post hoc test using SPSS Statistics (Version 29; IBM, Armonk, NY, USA). Statistical significance was set at *p* < 0.05.

## 5. Conclusions

While NS is a commonly utilized irrigation solution during arthroscopy, our data suggest that even brief exposure to unbuffered NS profoundly disrupts critical articular cartilage chondrocyte metabolism and synovial cell function largely due to the acidity of the solution. If this occurs at a large scale in a joint, it could adversely affect recovery from arthroscopic surgery and possibly joint health. Although the long-term clinical impact of this cytotoxicity is uncertain, it could adversely affect cellular recovery and joint health following arthroscopic surgery. However, it appears that these deleterious effects can be corrected by utilizing HEPES, a readily available biologic solution in order to minimize potential adverse effects of arthroscopic surgery.

## Figures and Tables

**Figure 1 ijms-25-01286-f001:**
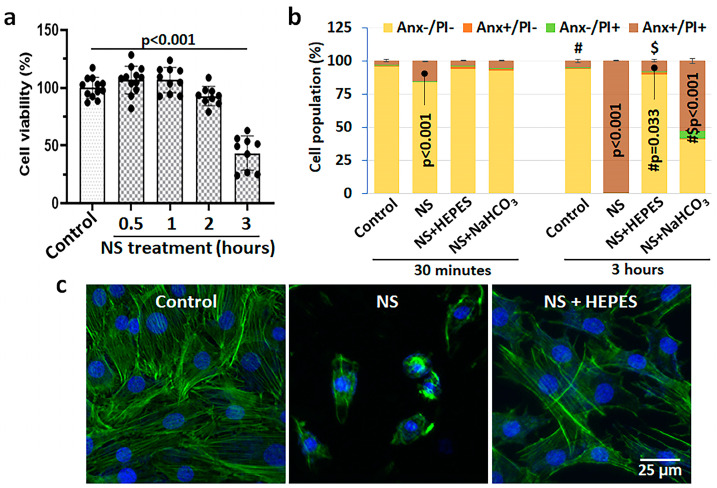
Cellular responses of various irrigants in monolayer primary bovine chondrocyte culture. (**a**) Cell viability according to the incubation time of 0.9% normal saline (NS) (*n* = 9–12). Data were normalized by control (culture media). (**b**) Population of chondrocyte death using a flow cytometer (*n* = 3–5). The cells were treated with culture medium, NS, or NS supplemented with 25 mM HEPES (N-2-hydroxyethylpiperazine-N′-2-ethanesulfonic acid) or 0.5 mM sodium bicarbonate (NaHCO_3_), and then stained with Annexin V (Anx) and propidium iodide (PI). *p*-values were presented from the population of Anx+/PI+ (#: control, $: NS+HEPES). (**c**) Representative images of fluoroprobe conjugated phalloidin immunofluorescence to visualize cytoskeletal changes of chondrocytes. The cells were exposed to culture medium, NS, or NS supplemented with 25 mM HEPES for 3 h. Green: Alexa Flour-488-Phalloidin (actin); Blue: DAPI (nucleus).

**Figure 2 ijms-25-01286-f002:**
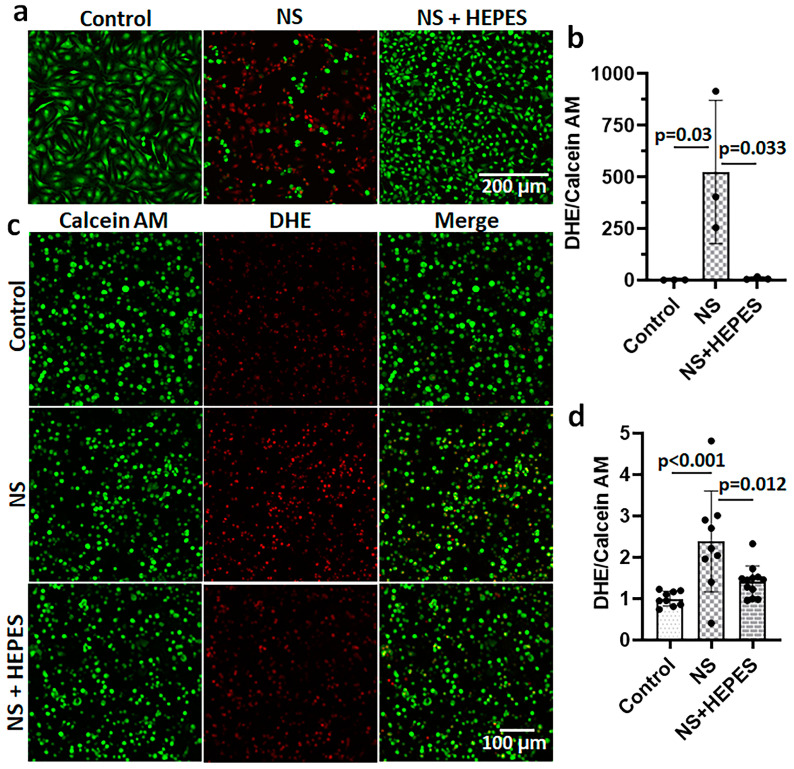
Effect of normal saline (NS) and HEPES (N-2-hydroxyethylpiperazine-N′-2-ethanesulfonic acid) supplementation on oxidative stress in monolayer primary bovine chondrocytes and agarose-based hydrogel culture systems. (**a**) Representative merged images of dihydroethidium (DHE; red)/Calcein AM (green) staining in chondrocyte monolayer culture after incubating with culture media (control), 0.9% NS, or 25 mM HEPES supplemented NS for 3 h. (**b**) Quantified DHE/Calcein AM ratio normalized by control (*n* = 3). (**c**) Representative individual images of dihydroethidium (DHE; red) and Calcein AM (green) staining in 3-dimensional chondrocytes/agarose culture after treating culture media (control) or NS for 3 h. (**d**) Quantified DHE/Calcein AM ratio normalized by control (*n* = 9).

**Figure 3 ijms-25-01286-f003:**
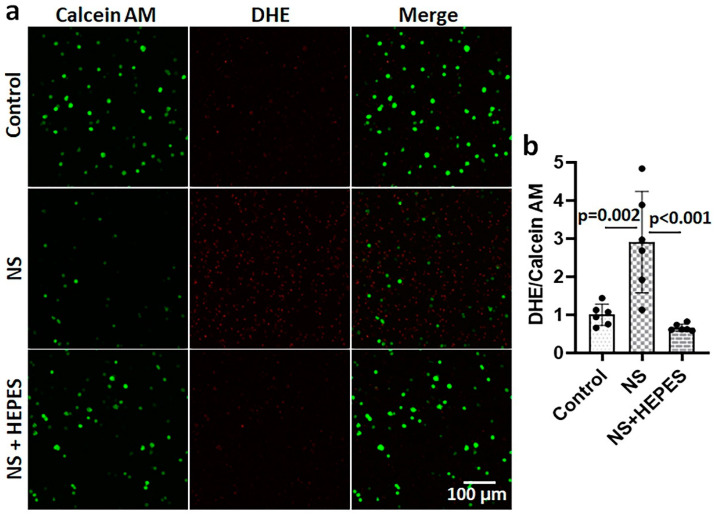
Effect of normal saline (NS) and HEPES supplementation on oxidative stress in primary bovine synoviocytes. (**a**) Dihydroethidium (DHE) staining in 3-dimensional synoviocytes/agarose culture after incubating culture medium (control), NS, or HEPES for 3 h. Red: DHE, Green: Calcein AM. (**b**) Quantified DHE/Calcein AM ratio normalized by control (culture media) (*n* = 6).

**Figure 4 ijms-25-01286-f004:**
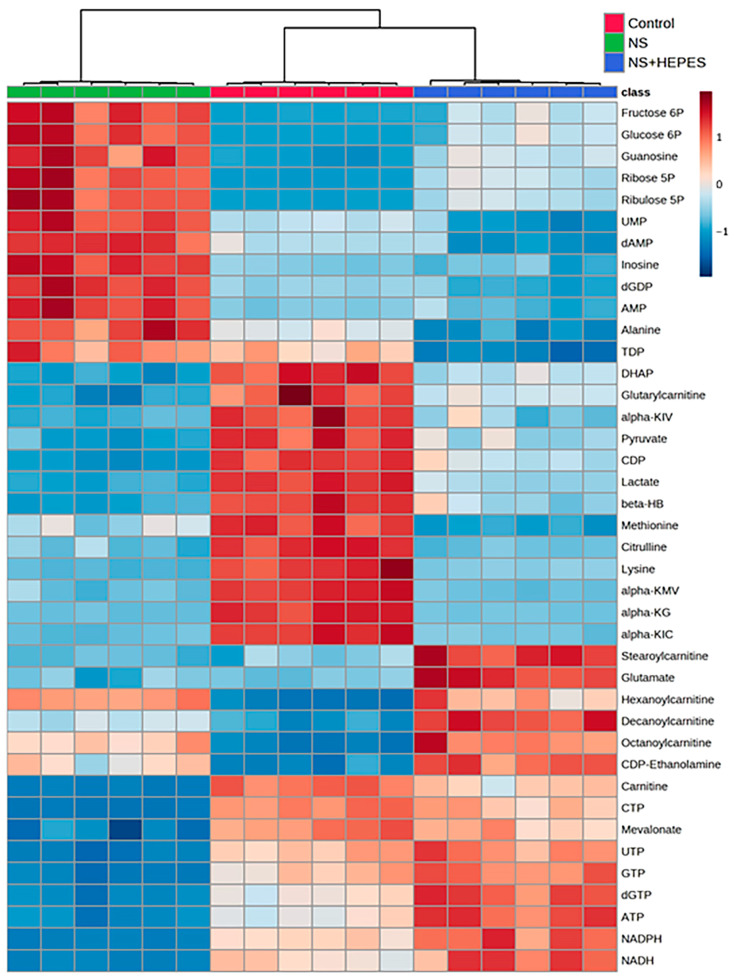
Heatmap with hierarchical clustering of metabolomics data. The top 40 metabolites were selected based on fold changes between control and 0.9% normal saline (NS). Chondrocytes were treated with each irrigation solution for 30 min (*n* = 6).

**Figure 5 ijms-25-01286-f005:**
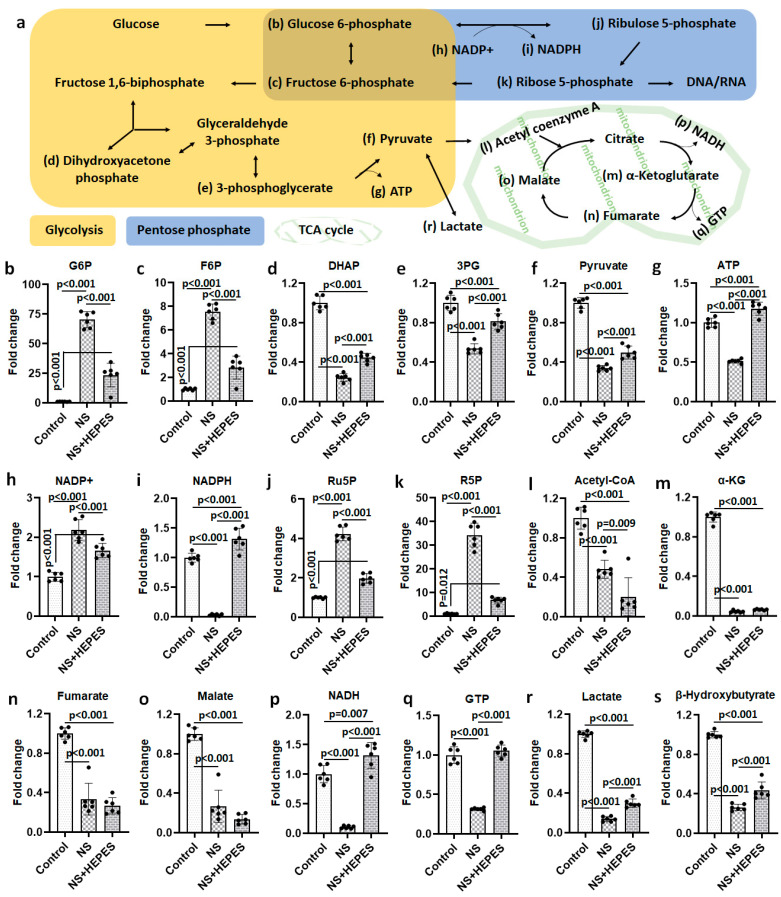
Metabolites related to strongly associated pathways. Chondrocytes were treated with each irrigation solution for 30 min (*n* = 6). (**a**) Glycolysis, pentose phosphate, and tricarboxylic acid (TCA) cycle pathways. (**b**–**s**) Metabolites normalized by the mean value of control: (**b**) Glucose 6-phosphate (G6P), (**c**) Fructose 6-phosphate (F6P), (**d**) Dihydroxyacetone phosphate (DHAP), (**e**) 3-phosphoglycerate (3PG), (**f**) Pyruvate, (**g**) Adenosine triphosphate (ATP) (**h**) Nicotinamide adenine dinucleotide phosphate (NADP+), (**i**) Nicotinamide adenine dinucleotide phosphate + hydrogen (NADPH), (**j**) Ribulose 5-phosphate (Ru5P), (**k**) Ribose 5-phosphate (R5P), (**l**) acetyl coenzyme A (Acetyl-CoA), (**m**) α-Ketoglutarate (α-KG), (**n**) Fumarate, (**o**) Malate, (**p**) Nicotinamide adenine dinucleotide + hydrogen (NADH), (**q**) Guanosine triphosphate (GTP), (**r**) Lactate, and (**s**) β-Hydroxybutyrate. NS: 0.9% normal saline.

**Table 1 ijms-25-01286-t001:** pH measurements of irrigation solutions.

Mean ± Standard Deviation	Control (Culture Media)	0.9% Normal Saline (NS)	25 mM HEPES in NS	0.5 mM NaHCO_3_ in NS
pH of solution (*n* = 3)	7.85 ± 0.009	5.30 ± 0.017	7.05 ± 0.005	7.30 ± 0.012
pH of solution after 30 min exposure with chondrocytes (*n* = 3)	7.84 ± 0.014	4.96 ± 0.026	6.87 ± 0.022	6.50 ± 0.022
pH of solution after 3 h exposure with chondrocytes (*n* = 3)	7.88 ± 0.034	4.79 ± 0.024	6.84 ± 0.017	6.55 ± 0.009

## Data Availability

The datasets generated during and/or analyzed during the current study are available from the corresponding author on reasonable request.

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
