# Peer review of "Buffering Mitigates Chondrocyte Oxidative Stress, Metabolic Dysfunction, and Death Induced by Normal Saline: Formulation of a Novel Arthroscopic Irrigant"

_ijms, 2024, doi:10.3390/ijms25021286_

Round 1

Reviewer 1 Report (Previous Reviewer 1)

Comments and Suggestions for Authors

I was only wondering why the intercellular sodium was reduced after sodium bicarbonat treatment.

Why is intracellular sodium reduced after the addition of sodium bicarbonate?

Author Response

I was only wondering why the intercellular sodium was reduced after sodium bicarbonate treatment. Why is intracellular sodium reduced after the addition of sodium bicarbonate?

There was no change of intracellular sodium concentration (ISC) after 30-minutes of NS exposure. However, the addition of 0.5 µM sodium bicarbonate (NaHCO3) in 0.9%(w/v) reduced ISC by approximately 15%. Although the reason for the drop is unclear, it has been reported that HCO3- can cause intracellular acidosis (doi: 10.1155/2015/605830), which can disrupt sodium entry into cells. Another possibility is that HCO3- inhibited uptake of the sodium probe. More studies are needed to clarify the underlying mechanisms mediating HCO3- effects. Given the uncertainties in interpreting these data we prefer to not discuss the findings in the manuscript.

Reviewer 2 Report (New Reviewer)

Comments and Suggestions for Authors

General information about the article:

The article Buffering Mitigates Chondrocyte Oxidative Stress, Metabolic Dysfunction, and Death Induced by Normal Saline: Formulation of a Novel Arthroscopic Irrigant addresses a critical concern in the field of arthroscopic procedures. The authors aim to investigate the detrimental effects of using 0.9% normal saline (NS) as a joint irrigant and propose a novel solution to mitigate its adverse impact on chondrocyte viability and metabolic function.

The introduction provides a concise background, highlighting the long-standing use of NS for joint irrigation despite emerging evidence of its negative impact on chondrocyte metabolism and articular cartilage function. The hypothesis that chondrocyte oxidative stress induced by low pH is the primary driver of NS toxicity is presented, setting the stage for the experimental investigation.

The methodology employed in the study is robust, with a focus on assessing chondrocyte viability, reactive oxygen species (ROS) production, and overall metabolic function. The findings reveal that even brief exposure to NS results in cell death, ROS overproduction, and disruption of essential metabolic pathways in chondrocytes. Furthermore, NS induces ROS overproduction in synovial cells, potentially affecting overall joint health.

The key innovation proposed in the article is the buffering of NS with 25 mM 4-(2-hydroxyethyl)-1-piperazineethanesulfonic acid (HEPES). The results demonstrate that this buffering significantly improves chondrocyte viability, reduces ROS production, and restores metabolite levels to near control levels. Additionally, the buffering agent proves effective in reducing ROS production in synovial cells. These findings strongly support the authors' hypothesis that the acidic pH of NS is a major contributor to its harmful effects and that buffering can effectively mitigate these adverse outcomes.

The conclusion effectively summarizes the study's major findings and emphasizes the importance of considering alternative formulations for arthroscopic irrigants to safeguard chondrocyte and synoviocyte health. The article not only contributes valuable insights to the scientific community but also suggests a practical solution to enhance the safety and efficacy of arthroscopic procedures.

Overall it is a well written paper, addressing important issue of adverse effects of saline being utilized in arthroscopic surgery. Simple regulation of pH by adding HEPES shows results which can’t be omitted in clinical practice. The only comment to this paper is that figure 1 is to early in the results section, therefore while reading it gets a little bit challenging to fully understand its nature. I would suggest that figures were firstly explained in the text, and after written explanation the graphical presentation of the results should appear. Overall I congratulate the authors of their work. 

Author Response

Thank you for your positive review. Figure 1 is moved after the texts.

This manuscript is a resubmission of an earlier submission. The following is a list of the peer review reports and author responses from that submission.

Round 1

Reviewer 1 Report

Comments and Suggestions for Authors

In the study, the authors investigate the effect of 0.9% saline and a mixture with HEPES on chondrocyte survival and metabolism. My main criticism of the study is the design. I find it surprising that pure saline was placed on the cells. With this in mind, I am not surprised that the survival of the chondrocytes is limited after only a short time. If NaCl was added to the cell culture medium, the situation would probably be different. Synovial fibroblasts are not affected by additional saline. To really conclude on pH as a cause for the differences, decisive experiments are missing. In addition, the sodium concentration in the solution and in the cells should be measured.

Reviewer 2 Report

Comments and Suggestions for Authors

Arman and colleagues investigated the impacts of saline on chondrocytes, identifying some detrimental effects of 0.9% normal saline (NS) which could potentially be mitigated by the application of HEPES. This research bears clinical significance, particularly in light of the common use of 0.9% NS during arthroscopy procedures. While the study is indeed intriguing, it could benefit from addressing the following points to further enhance its contributions:

1. It would augment the research if the authors could incorporate FACS results for apoptosis in Figure 1 to provide a deeper understanding of the phenomena being investigated.

2. In Figure 3A, the presentation of fluorescence results for the control, NS, and NS+HEPES groups is essential to offer a comprehensive view of the experiment’s outcomes.

3. The choice of HEPES to counteract the adverse effects of NS on chondrocytes is central to your study. Elaborating on the rationale behind selecting HEPES, both in the introduction and discussion sections, would strengthen the paper by providing necessary context and background.

4. The hypothesis presented, that HEPES principally shields against the harmful effects by modulating the pH of the saline solution, warrants further exploration. Are there additional mechanisms through which HEPES could exert protective effects? Furthermore, if pH adjustment is indeed the critical factor, testing other substances that can normalize the saline's pH might affirm this hypothesis and broaden the study's scope. Investigating this avenue could potentially lend more weight to your conclusions.

Comments on the Quality of English Language

Overall, the English language quality of this passage is quite good, showcasing a high level of technical knowledge and adherence to the conventions of scientific writing. Some minor adjustments to sentence structure and word choice, as well as correcting the formatting errors, could enhance the readability and clarity of the passage.